# A study protocol to investigate if acipimox improves muscle function and sarcopenia: an open-label, uncontrolled, before-and-after experimental medicine feasibility study in community-dwelling older adults

Claire McDonald [1,2,3] Craig Alderson,[4] Matthew G Birkbeck,[4,5] Laura Brown,[6] Silvia Del Din,[7] Grainne G Gorman,[6] Kieren Hollingsworth,[5] Clare Massarella,[4] Rana Rehman,[7] Lynn Rochester,[2,4,7] Avan AP Sayer,[1,2] Huizhong Su,[7] Helen Tuppen,[6] Charlotte Warren,[6] Miles D Witham [1,2]

For numbered affiliations see end of article.

**Correspondence to**
Dr Miles D Witham;
miles.witham@newcastle.ac.uk

## ABSTRACT

**Introduction** Sarcopenia is the age-associated loss of muscle mass and strength. Nicotinamide adenine dinucleotide (NAD) plays a central role in both mitochondrial function and cellular ageing processes implicated in sarcopenia. NAD concentrations are low in older people with sarcopenia, and increasing skeletal muscle NAD concentrations may offer a novel therapy for this condition. Acipimox is a licensed lipid-lowering agent known to act as an NAD precursor. This open-label, uncontrolled, before-and-after proof-of-concept experimental medicine study will test whether daily supplementation with acipimox improves skeletal muscle NAD concentrations.

**Methods and analysis** Sixteen participants aged 65 and over with probable sarcopenia will receive acipimox 250 mg and aspirin 75 mg orally daily for 4 weeks, with the frequency of acipimox administration being dependent on renal function. Muscle biopsy of the vastus lateralis and MRI scanning of the lower leg will be performed at baseline before starting acipimox and after 3 weeks of treatment. Adverse events will be recorded for the duration of the trial. The primary outcome, analysed in a per-protocol population, is the change in skeletal muscle NAD concentration between baseline and follow-up. Secondary outcomes include changes in phosphocreatine recovery rate by [31]P magnetic resonance spectroscopy, changes in physical performance and daily activity (handgrip strength, 4 m walk and 7-day accelerometry), changes in skeletal muscle mitochondrial respiratory function, changes in skeletal muscle mitochondrial DNA copy number and changes in NAD concentrations in whole blood as a putative biomarker for future participant selection.

**Ethics and dissemination** The trial is approved by the UK Medicines and Healthcare Products Regulatory Agency (EuDRACT 2021-000993-28) and UK Health Research Authority and Northeast – Tyne and Wear South Research Ethics Committee (IRAS 293565). Results will be made available to participants, their families, patients with sarcopenia, the public, regional and national clinical teams, and the international scientific community.
**Protocol** Acipimox feasibility study Clinical Trial Protocol V.2 2/11/21.
**Trial registration number** The ISRCTN trial database (ISRCTN87404878).

## STRENGTHS AND LIMITATIONS OF THIS STUDY

⇒ A key strength is that our recruitment method targets older people with sarcopenia—a population traditionally hard to include in trials and with whom experimental medicine studies are not commonly performed.

⇒ Our use of muscle biopsy-based outcome measures to understand treatment mechanisms is another key methodological strength and will demonstrate the feasibility of doing serial muscle biopsies in this population for future similar trials.

⇒ A limitation is that the study is not randomised, and larger randomised controlled trials will be required to assess efficacy if results from this proof-of-concept trial are promising.

⇒ Coadministration of aspirin with acipimox is a methodological limitation as this complicates efforts to understand the mechanism of any beneficial effect, although it should improve the tolerability and hence adherence to the treatment.

## INTRODUCTION

Sarcopenia is the loss of muscle strength and mass that commonly accompanies ageing. It is a syndrome of major clinical importance for many older people and affects between 5% and 10% of those aged over 65 in the general population, with rates in excess of 30% among those living in care homes.[1 2]

Sarcopenia is associated with an increased risk of falls, hospitalisation, need for care and earlier death.[3–5] The cost of sarcopenia to the UK health service has been estimated to be in the region of 2 billion pounds per year.[6]

In recent decades, several definitions of sarcopenia have been proposed and there is not yet global consensus on a single definition or operationalisation. The revised European consensus on the definition and diagnosis of sarcopenia requires low muscle strength (demonstrated by low handgrip strength or prolonged sit-to-stand time) to make a diagnosis of probable sarcopenia; low muscle mass can optionally be used to confirm the diagnosis.[7] At present, resistance exercise is the only intervention shown to be of benefit in sarcopenia.[2] However, this intervention is only partially effective at reversing the decline in muscle mass and strength, and not all older people are either willing or able to undertake resistance training. Alternative interventions are therefore required both to prevent and ameliorate sarcopenia and its deleterious consequences.

The pathophysiology of sarcopenia is complex, multifactorial and incompletely understood. Impaired microvascular and macrovascular function, neuromuscular junction dysfunction, chronic inflammation and changes to the extracellular matrix have all been posited as being of importance.[7] Mitochondrial dysfunction, however, is postulated to play a central role in the aetiology of the condition. Mitochondrial numbers are low in sarcopenic muscle, and oxidative phosphorylation capacity is strikingly impaired, driven by reductions in expression and activity of all the major mitochondrial respiratory chain complexes.[8 9] Ex vivo mitochondrial oxidative capacity parallels decline in muscle strength.[10] Furthermore, release of proapoptotic factors, morphological alterations (fission, swelling), energy stress via reduced ATP and increased mitochondrial reactive oxygen species emission (which react with and damage cellular components including contractile proteins) have all been reported during muscle atrophy in preclinical studies.[10] Interventions that target deficits in mitochondrial function therefore provide an attractive mechanism to treat sarcopenia.

## The role of NAD+ in muscle function

NAD+ and its reduced form (NADH) plays a central role in cellular redox homeostasis and is a cofactor in multiple pathways of cellular metabolism. NADH is essential for ATP production via oxidative phosphorylation in mitochondria, but it is also now clear that NAD+ is a key signalling molecule for multiple pathways implicated in cellular ageing.[11] These include roles in protein deacetylation signalling via sirtuins and DNA damage sensing (via poly-ADP-ribose polymerases; PARPs). These and other key NAD+-dependent systems such as the NADase CD38, which increases with ageing, not only play important roles in cellular ageing, but they also consume NAD+, leaving less for competing systems. NAD+ concentrations are lower at older age in multiple tissues, and lower concentrations are found in a wide range of degenerative

diseases.[12] Recent data confirm that NAD+ concentrations are lower (by 32%) in muscle biopsies from older people with sarcopenia than in healthy, age-matched controls.[13] Therapies that increase NAD+ concentrations may therefore provide a way to reverse the changes commonly seen in ageing skeletal muscle which contribute to sarcopenia, including deficits in mitochondrial function.[11 12]

Acipimox is a nicotinic acid analogue and postulated to act as an NAD+ precursor.[14] Administration of acipimox may therefore be able to enhance skeletal muscle NAD concentrations in patients with sarcopenia, with consequent improvement in mitochondrial function and hence skeletal muscle function. Acipimox already has marketing authorisation for use as a lipid-lowering agent for patients with type IIb or type IV hyperlipidaemia who have not responded to other therapies. A recent systematic review identified 23 randomised controlled trials assessing the effect of NAD precursors on measures of physical function.[11] Studies were small (2–8 participants). A total of 96 primary outcomes were assessed, 10 of which were in favour of an NAD precursor and 1 was in favour of placebo; the remainder were not statistically significant in any clear direction. Small studies of acipimox that enrolled healthy younger patients or those with left ventricular dysfunction did not show improvement in measures of physical function.[15–19]

In contrast, muscle biopsy data from a trial of 21 patients with type 2 diabetes mellitus showed that 2 weeks of acipimox treatment (250 mg two times per day) produced significant upregulation of mitochondrial genes involved in oxidative phosphorylation, an increase in skeletal muscle ATP content and an increase in ex vivo mitochondrial respiration; no change in mitochondrial number measured by mtDNA copy number was noted.[14] The results of a placebo-controlled trial examining the impact of acipimox on mitochondrial function in patients with mitochondrial myopathy are awaited.[20] However, none of these studies have assessed the presence of sarcopenia or frailty and enrolled younger or middle-aged participants.

### Trial objectives

Trials to date have not tested acipimox or other NAD analogues in older people with sarcopenia or frailty. Testing whether acipimox has the expected biological effects in this population (which would be the target population for treatment in clinical practice) is a key step in ascertaining whether this therapy should be tested in larger randomised controlled trials for people with sarcopenia and frailty. The primary objective of this proof-of-concept experimental medicine study is therefore to test whether supplementation with acipimox improves skeletal muscle NAD concentrations. The secondary objectives are to test whether supplementation with acipimox improves mitochondrial respiratory chain function, to examine if individuals with low NAD concentrations in skeletal muscle biopsy can be identified by less invasive biomarkers (whole blood NAD concentration) and to

assess candidate outcomes for future randomised clinical trials in patients with sarcopenia and frailty.

## METHODS AND ANALYSIS

### Trial design and setting

This is an open-label, uncontrolled, before-and-after feasibility study. Participants will be approached through older people's medicine clinics, day units and rehabilitation units. In addition, potential participants will be identified through local primary care centres, acting as patient identification centres, and via a sarcopenia registry.[21] Screening visits will take place either in the participants' homes or in a clinical research facility. Study assessments will take place at the Clinical Ageing Research Unit and Clinical Research Facilities, based in Newcastle upon Tyne Hospitals NHS Foundation Trust.

### Eligibility criteria

The target population are adults aged 65 years and over with sarcopenia. Sarcopenia is defined using handgrip strength or five times sit-to-stand thresholds from the 2019 European Working Group on Sarcopenia in Older People guidance.[7] In order to identify a frail/prefrail population, participants will also be required to have a walk speed <=0.8 m/s on a 4-metre walk test.[22] Participants must be able to give written informed consent. The study exclusion criteria are primarily focused on excluding participants for whom use of acipimox or aspirin may be unsafe or for whom MRI is contraindicated. Participants who are at risk of other types of skeletal muscle myopathy will be excluded to avoid confusion with the diagnosis of sarcopenia. The full inclusion and exclusion criteria for the study are shown in Box 1.

### Interventions

All participants will take acipimox orally. Participants will be supplied with 250 mg capsules, and the dose is adjusted according to renal function. Participants with creatinine clearance (estimated by the Cockcroft-Gault equation) >60 mL/min will be prescribed one capsule three times per day.[23] Participants with creatinine clearance of 45–60 mL/min will be prescribed one capsule two times per day. Commonly reported adverse effects of acipimox are headache, dyspepsia, abdominal discomfort, urticaria, asthenia and facial flushing. Facial flushing is likely to increase the risk of discontinuation of therapy and to cause unmasking of treatment allocation in any future randomised controlled trial. To suppress this side effect, participants will be prescribed aspirin 75 mg orally to be taken 30 min before the first dose of acipimox each day with the expectation that this intervention would also form part of a future randomised controlled trial as has been the case previously.[20] Participants who take aspirin as part of their usual medication regime will continue their usual dose and will not be prescribed additional aspirin. If acipimox is discontinued, aspirin will be discontinued. Participants will start acipimox and aspirin 48 hours after

---

**Box 1  Inclusion and exclusion criteria**

Inclusion criteria
Age 65 years or over
Low maximum handgrip strength (<16 kg for women, <27 kg for men) OR five times sit-to-stand time >15 s (inability to complete sit-to-stand test will count as time >15 s)
Walk speed <=0.8 m/s on 4-metre walk test
Exclusion criteria
General:
Allergy to acipimox or other niacin-related products
Allergy or intolerance of aspirin
Any contraindication to taking aspirin
Unable to give written informed consent
Currently enrolled in another intervention study (observational studies are permitted)
Currently participating in supervised exercise classes or physiotherapy
Any progressive neurological or malignant condition with life expectancy <6 months
Safety of investigations medications:
Creatinine clearance <45 mL/min (by Cockcroft-Gault equation)
Taking statin medication or fibrate medication
Active peptic ulcer disease or dyspepsia
Safety of muscle biopsy and MRI:
Platelets <100×10$^9$/L at screening (contraindication to muscle biopsy)
Presence of a bleeding diathesis or use of oral or parenteral anticoagulant medication
Antiplatelet agents other than low dose (75 mg one time per day) aspirin
Contraindications to MRI scanning (mild claustrophobia is not a contraindication)
Allergy to local anaesthetic (lidocaine)
Unable to palpate vastus lateralis muscle to enable biopsy localisation
Other causes of skeletal myopathy:
Liver function tests (bilirubin, alanine transaminase, alkaline phosphatase) > 3 × upper limit of normal
Symptomatic (New York Heart Association (NYHA) class II–IV) chronic heart failure (diagnosed according to European Society of Cardiology guidelines)
Severe Chronic obstructive pulmonary disease (COPD) (GOLD stage IV)
Known myositis or other established myopathy
Self-reported weight loss of >10% in the last 6 months (to exclude significant cachexia)
Known uncontrolled thyrotoxicosis
7.5 mg/day or greater prednisolone use (or equivalent)

---

the initial muscle biopsy is completed and continue the intervention until 24 hours prior to the second biopsy (3 weeks±3 days). This dose duration of treatment was sufficient to upregulate mitochondrial genes involved in oxidative phosphorylation, increase in skeletal muscle ATP content and increase ex vivo mitochondrial respiration in a study of patients with type 2 diabetes mellitus.[14]

Investigators may discontinue the trial treatment in the event of intolerable side effects occurring that are possibly, probably or definitely related to trial medication or which constitute a serious adverse reaction (SAR) or suspected unexpected serious adverse reaction (SUSAR). Treatment will also be discontinued if the participant requests the medication to be withdrawn, requests not to undergo follow-up study measurements or if creatinine

clearance falls below 45 mL/min. If creatinine clearance falls from above 60 mL/min to between 45 and 60 mL/min during the study, the dose frequency will be reduced from three times per day to two times per day. If acipimox is discontinued, this will automatically lead to withdrawal from the trial.

### Adherence

Participants will be asked to bring their unused trial medication to the follow-up visit so that adherence can be calculated by tablet count. Adherence will be defined as number of tablets taken divided by number of tablets expected to have been taken.

### Outcomes

#### Primary outcome

The primary outcome is change in skeletal muscle NAD+/NADH concentrations and ratio between baseline and follow-up. To measure this outcome, muscle biopsy of vastus lateralis muscle will be performed at baseline and follow-up. Biopsies will be taken under local anaesthetic, using a Weil-Blakesley conchotome, a technique that has previously been shown to be well tolerated even by very old participants.[24–26] The second biopsy will be taken from the same leg as the first biopsy, a minimum of 3 cm from the first biopsy site to avoid healing or inflammatory changes affecting the results of the second biopsy. Participants will be asked to withhold aspirin for 24 hours prior to biopsy. Aspirin (taken as part of their usual medication regime or as adjunctive medication in the study to prevent flushing) can be started or restarted 48 hours after the biopsy. Biopsy samples will be weighed and snap frozen in liquid nitrogen cooled isopentane immediately after extraction, then stored at −80°C pending analysis. NAD+/NADH concentrations will be measured using a modified protocol of the Promega NAD/NADH-Glo assay kit (Promega, Southampton, UK).

#### Secondary outcomes

A range of measures of physical performance, mechanistic outcomes and outcomes related to trial performance will also be recorded at baseline and follow-up to assess the potential of these outcomes for future randomised controlled trials.

#### Biopsy-related outcomes

ATP/ADP concentrations and ratio will be assessed in the muscle biopsy samples according to Strehler;[27] cytochrome C oxidase/ succinate dyhydrogenase (COX SDH) histochemistry[28] and quadruple immunofluorescence[29] will be used to assess respiratory chain deficiency. Mitochondrial DNA copy number will be assessed via quantitative PCR.[30]

#### Physical performance measures

Maximal handgrip strength (measured using a Jamar dynamometer) and the short physical performance battery will be performed at baseline and follow-up.[31 32]

#### Remote digital monitoring

Seven-day remote monitoring with a wearable device (AX6; Axivity, Newcastle upon Tyne, UK) will be used to measure indices of walking activity, gait speed, variability and postural control at baseline and follow-up.[33–35]

#### Magnetic resonance imaging and magnetic resonance spectroscopy outcomes

MR spectroscopy will assess phosphocreatine recovery rate measured by 31P magnetic resonance spectroscopy (MRS) of the calf. The full MRI protocol has been published previously.[36] Changes in NAD(P)H levels by 31P MRS will also be assessed as an exploratory outcome. MRI of the calf muscle will be conducted at baseline and follow-up. Quantitative Dixon MRI will assess for intramuscular fat infiltration.[37 38] An exploratory, diffusion-weighted MRI technique—motor unit MRI—will be performed at rest to assess motor unit size via spontaneous muscle activity.[39]

#### Whole blood NAD concentration

Whole blood will be collected into EDTA tubes and frozen at −80°C. Samples will be analysed for NAD+/NADH concentrations and ratio using Q-NADMED Blood NAD+ and NADH assay kit (NADMED, Helsinki, Finland) to test whether a precision target population (individuals with low skeletal muscle NAD+/NADH ratios at biopsy) can be identified by less-invasive biomarkers.

### Participation timeline

#### Recruitment and data management

Recruitment will take place over 12 months and started in October 2022. Potential participants are sent or given a brief study information sheet with an invitation letter, a reply slip and a prepaid envelope to express interest. The reply slip is returned to the study team and those who are interested will be contacted for prescreening as described below. Data including numbers of potential participants identified, approached and prescreened will be collected and documented on the site screening log. Data will be collected on a paper case report forms and transposed to and secure Excel database. The participant trial record, including completed paper data collection tools, will be archived at site for 15 years following the end of the trial.

#### Prescreening

Participants who express an interest in the trial will enter a brief telephone prescreening process to check provisional eligibility. The prescreening form is shown (online supplemental material 1). Verbal consent will be taken for the prescreening process including access to medical records. The SARC-F questionnaire will be administered[40]; this comprises five questions on physical function giving a total score between 0 and 10. Based on previous observational data and trial experience, a score of one or more will be sufficient to identify those more likely to have sarcopenia and will enable progression to the screening visit.[15 41] Participants who pass the prescreen will be given, emailed or posted the full participant information sheet (PIS), and all participants will be given at least 48 hours

to consider their participation. The PIS can be found in online supplemental material 2. The study team will then telephone the potential participant, and those wishing to take part will then be invited to a screening visit.

### Screening

Written informed consent will be obtained at the screening visit, which may take place at home, in clinic or in a research facility. After informed consent is given, information on demographics, medical history and medication history (including history of allergy to acipimox or niacin-related products) will be collected. Maximum handgrip, five times sit-to-stand and 4-metre walk speed will be measured.[22] Screening blood will be taken to ensure it is safe to prescribe acipimox and perform muscle biopsy. A single 4 mL EDTA sample will be collected for full blood count, and a single 8.5 mL clot activator separation gel sample will be collected for urea, creatinine and electrolytes and liver function tests (LFTs—bilirubin, alanine aminotransferase and alkaline phosphatase). Acceptable results are shown in Box 1. The vastus lateralis muscle will be inspected and palpated to ensure suitability for biopsy. Participants will be asked to confirm if they are taking part in any other interventional clinical trials or monitored exercise programmes. Participant eligibility to proceed to the baseline visit is confirmed after the screening visit, once all screening assessment results are available.

### Schedule of events

The schedule of events for the study is shown in online supplemental table 1. The primary and secondary outcomes are measured over two baseline visits and repeated at two follow-up visits.

### Sample size

As this is a feasibility study, few data exist on which to base a sample size calculation. The sample size has been selected to enable detection of a one SD change in measures of mitochondrial NAD+/NADH ratios, ATP/ADP ratios or respiratory chain function. To detect this change with an alpha of 0.05 and 80% power requires 11 paired observations (using a paired t-test). We plan to recruit 16 participants to allow for dropout and non-completion of the course of medication.

### Statistical analysis
#### Analysis population
The primary analysis population is the per-protocol population. This will consist of all participants who do not discontinue acipimox and undergo both baseline and follow-up assessment for a given outcome. A safety analysis will be performed examining adverse events for all participants receiving at least one dose of acipimox.

#### Statistical analysis plan
A full statistical analysis plan will be developed during the trial, which will include evaluation of the mechanistic outcome data and correlation with main study outcomes.

This will be developed by the Trial Management Group. Descriptive baseline statistics for all enrolled participants and for those completing the primary analysis (the per-protocol group) will be generated.

#### Analysis of the primary outcome measure
The primary outcome is change in skeletal muscle NAD+/NADH ratios between baseline and follow-up. Results will be analysed using a paired t-test (if normally distributed) or a Wilcoxon signed-rank test (if not normally distributed).

#### Analysis of secondary outcome measures
Continuous secondary endpoints will be analysed in a similar manner to the primary outcome. Binary secondary endpoints will be analysed with a logistic regression model. Correlation between the changes in skeletal muscle NAD+/NADH ratios and changes in selected biomarkers (phosphocreatine recovery rate by MR spectroscopy, NAD+/NADH ratios in blood) will be performed using Pearson's correlation coefficient (where measures are normally distributed) or Spearman's rho (where measures are not normally distributed).

### Trial oversight
A Trial Management Group (TMG) will convene approximately monthly throughout the trial. Members will include the Chief Investigator, co-applicants, trial statistician, local site research staff and a Sponsor representative. The TMG undertakes the role of a Trial Steering Committee for this small, low-risk Clinical Trial of an Investigational Medicinal Product study. No independent data monitoring committee will be established given the low number of participants and the short duration of therapy. Instead, an interim safety review by the funder will examine the data from the first eight patients to complete the study.

### Patient and public involvement
Patient consultation influenced the choice of outcomes. Patients told us that both muscle biopsy and 31P MRS scanning would be acceptable even in this older, frailer population. They supported an early phase feasibility study to confirm this and improve our mechanistic understanding of how acipimox may improve muscle function. Results from this study will be used to inform design process for a future randomised controlled trial, which would be powered for patient-centred outcomes such as physical performance and quality of life.

### Harms
All adverse events (AEs) occurring from point of consent to end of trial participation will be recorded. Falls will not be recorded as AEs, but any injury resulting from a fall will be recorded as an adverse event. Mild pain/soreness for the first 48 hours after muscle biopsy and superficial bruising at the site are to be expected and will not be recorded as AEs. Investigations, interventions, procedures and operations will not be recorded as AEs, but the

underlying condition (if new) leading to the intervention will be recorded as an AE. Complications resulting from investigations, interventions, procedures and operations will be recorded as AEs.

## Consent

Written informed consent will be obtained at screening visit before conducting any trial procedures including screening assessment. A medically qualified Investigator, competent in obtaining informed consent for research studies and formally delegated to do so by the site Principal Investigator, by way of dated signatures on the site delegation log, will conduct a consent interview with the participant at the clinic or in the participant's own home if preferred by the participant.

## Ancillary and post-trial care

No provision for continuation of trial medication will be made by the trial team or Sponsor. Acipimox is not licensed for the indication under study in this trial; any off-licence use of acipimox after the end of the trial would be the responsibility of the participant's usual primary or secondary care clinical team.

## ETHICS AND DISSEMINATION
### Ethics approval

A favourable ethical opinion has been granted from the UK Health Research Authority and Northeast – Tyne and Wear South Research Ethics Committee (IRAS 293565). The trial has also received approval from the UK Medicines and Healthcare Products Regulatory Agency (EuDRACT 2021-000993-28). The trial has been included in the National Institute for Health Research Clinical Research Network (NIHR CRN) portfolio (CPMS49429). The trial Sponsor is the Newcastle upon Tyne Hospitals NHS Foundation Trust, Freeman Hospital, Freeman Road, High Heaton, Newcastle upon Tyne, NE7 7DN. The trial Sponsor has delegated responsibility for trial management to the CI, including trial design; review and approval of all localised patient-facing documentation prior to implementation at each site; collection, analysis and interpretation of data; writing of the protocol publication and final clinical report manuscripts. This protocol manuscript is based on V.2.0 of the trial protocol (dated 2 November 2021).

### Dissemination policy

A final report of the trial will be provided to the Sponsor, Research Ethics Committee and the trial Funder within 1 year of the end of the study. The trial results will be uploaded to the European Union Drug Regulating Authorities Clinical Trials (EudraCT) database, as per the European Commission's guidelines on posting and publication of result-related information within 12 months. The trial is registered on the ISRCTN trial database (ISRCTN87404878) and trial results will be made publicly available on the ISRCTN trial registry within 12 months of the end of the trial, defined as Last Patient Last Visit date. The final clinical trial report will be used for publication and presentation at scientific meetings. Summaries of results will also be made available to investigators for dissemination within their clinical areas and to the wider public, and a summary of results will be sent to all participants along with their treatment allocation. We will hold a dissemination event for participants and their families to present and discuss the study results.

### Access to data

Access to the full dataset will be limited to the TMG and to authors of the trial publication. At the end of the trial, a deidentified dataset will be prepared and stored by Newcastle University. Requests for data sharing with study teams outside Newcastle University or the study Sponsor, including international requests, will be considered by a Data Access Committee with representation from the Funder, Sponsor and the trial Chief Investigator.

**Author affiliations**
[1]AGE Research Group, Translational and Clinical Research Institute, Faculty of Medical Sciences, Newcastle University, Newcastle upon Tyne, UK
[2]NIHR Newcastle Biomedical Research Centre, Newcastle Upon Tyne Hospitals NHS Foundation Trust, Newcastle Upon Tyne, UK
[3]Gateshead Health NHS Foundation trust, Gateshead, UK
[4]Newcastle Upon Tyne Hospitals NHS Foundation Trust, Newcastle Upon Tyne, UK
[5]Newcastle Magnetic Resonance Centre Translational and Clinical Research Institute, Newcastle University, Newcastle upon Tyne, UK
[6]Wellcome Centre for Mitochondrial Research, Translational and Clinical Research Institute, Newcastle University, Newcastle upon Tyne, UK
[7]Brain and Movement Research Group, Translational and Clinical Research Institute, Newcastle University, Newcastle upon Tyne, UK

**Acknowledgements** CMcD, MDW, AAS, GG, SDD and LR acknowledge support from the National Institute for Health and Care Research (NIHR) Newcastle Biomedical Research Centre (BRC). The NIHR Newcastle BRC is a partnership between Newcastle Hospitals NHS Foundation Trust, Cumbria Northumberland Tyne and Wear NHS Foundation Trust, and Newcastle University. The views expressed are those of the authors and not necessarily those of the NIHR or the Department of Health and Social Care.SDD and LR were also supported by the Mobilise-D project that has received funding from the Innovative Medicines Initiative 2 Joint Undertaking (JU) under grant agreement no. 820820. This JU receives support from the European Union's Horizon 2020 research and innovation program and the European Federation of Pharmaceutical Industries and Associations (EFPIA). SDD and LR were also supported by the Innovative Medicines Initiative 2 Joint Undertaking (IMI2 JU) project IDEA-FAST - Grant Agreement 853981. SDD and LR were supported by the NIHR/Wellcome Trust Clinical Research Facility (CRF) infrastructure at Newcastle upon Tyne Hospitals NHS Foundation Trust. We acknowledge the contribution of our much-missed colleague Dr Richard Dodds, who sadly died before this work could be completed.

**Contributors** Overall study conception and design: MDW, AAPS, KH and GGG; LRMRI design: MGB and KH. Laboratory analysis design and processes: LB, GGG, CW, HT, HS and CM. Accelerometry data collection and analysis design: SDD, RR and LR. Recruitment and medication use design: CA and CMcD.

**Funding** This work was funded by the UKRI Medical Research Council Confidence in Concept funding scheme (grant number MC_PC_19047).

**Competing interests** S. Del Din reports consultancy activity with Hoffmann-La Roche Ltd. outside of this study.

**Patient and public involvement** Patients and/or the public were involved in the design, or conduct, or reporting, or dissemination plans of this research. Refer to the Methods section for further details.

**Patient consent for publication** Not applicable.

**Provenance and peer review** Not commissioned; externally peer reviewed.

**ORCID iDs**
Claire McDonald http://orcid.org/0000-0002-7592-9952
Miles D Witham http://orcid.org/0000-0002-1967-0990

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
