## [Reviewer comments · BMJ Open]

ARTICLE DETAILS

TITLE (PROVISIONAL)	A study protocol to investigate if Acipimox improves muscle function and sarcopenia –an open-label, uncontrolled, before-and-after experimental medicine feasibility study in community dwelling older adults.
AUTHORS	McDonald, Claire; Alderson, Craig; Birkbeck, Matthew; Brown, Laura; Del Din, Silvia; Gorman, Grainne; Hollingsworth, Kieren; Massarella, Clare; Rehman, Rana; Rochester, Lynn; Sayer, Avan; Su, Huizhong; Tuppen, Helen; Warren, Charlotte; Witham, Miles

VERSION 1 – REVIEW

REVIEWER	Chilibeck, Philip University of Saskatchewan
REVIEW RETURNED	05-Sep-2023

GENERAL COMMENTS	The manuscript outlines a protocol involving acipimox treatment for improving muscle NAD+ content in older adults with sarcopenia and frailty. Introduction: I suggest providing details on how sarcopenia is measured in the introduction. There is controversy on the most adequate way to measure this. Mitochondrial dysfunction is associated with sarcopenia. Could the authors better describe the link in the introduction – is it due to excessive production of reactive oxygen species which induce damage to cellular membranes? What are the usual adverse events associated with acipimox? Page 8, Interventions: Please include the length of the intervention in this section. I believe this is three weeks, according to the abstract. Please provide justification for this length of intervention. Is this long enough to show significant changes in your primary and secondary outcomes? Is there any chance that co-administration of aspirin might affect some of your outcomes as NSAIDs might affect muscle mass and strength (please see: https://pubmed.ncbi.nlm.nih.gov/21160058/)? Page 12: Sample size: It is indicated that a projected change of 1 SD was used in the sample size calculation. Please indicate the exact number that was used in the sample size calculation (i.e., the actual NAD+/NADH that corresponds to 1 SD). Page 14: Were patient-desired outcomes actually the biopsy and P-MRS outcomes? I would suspect patients might not understand
---

	mechanistic outcomes and might place more importance on functional outcomes (i.e., strength, gait speed).
--	---

REVIEWER	Barbosa-Cortés, Lourdes Instituto Mexicano del Seguro Social Unidad de Investigación Médica en Nutrición, Unidad de Investigación Médica en Nutrición
REVIEW RETURNED	14-Oct-2023

GENERAL COMMENTS	1. Is the research question or study objective clearly defined? Yes, I consider that the study objective is clearly defined. In this open-label, uncontrolled, before-and-after experimental medicine study. The authors will test whether daily supplementation with acipimox improves skeletal muscle NAD concentrations. Besides, the authors present a critical, comprehensive, and up-to-date background review that supports the objective of the study. 2. Is the abstract accurate, balanced, and complete? The abstract is a synthesized version of the protocol, it contains the rationale for the research, the main objective, the design, and the methodology. In general, the summary is balanced. 3. Is the study design appropriate to answer the research question? Yes, I consider that this open-label, uncontrolled, before-and-after proof-of-concept experimental medicine study is appropriate to answer if the daily supplementation with acipimox improves skeletal muscle NAD concentrations established by the authors. 4. Are the methods described sufficiently to allow the study to be repeated? Yes, the authors describe in detail the procedures to be followed in order to meet the objectives and ensure the reproducibility of the study. For example, the authors describe the population of the study (old people, > 60 years and with sarcopenia), and the procedures to be followed in the execution of the objectives clearly. 1. Primary outcome. To measure change in skeletal muscle NAD⁺/NADH concentrations and ratio between baseline and follow up, the muscle biopsy of the vastus lateralis muscle will be performed at baseline and at 3 weeks after administration of acipimox. The authors mention the Biopsies will be taken under local anesthesia, using a Weil-Blakesley conchotome, a technique that has been shown to be well tolerated even by very elderly participants. Besides, participants will be asked to withhold aspirin for 24 hours prior the biopsy. Biopsy samples will be weighed and snap-frozen in liquid nitrogen cooled then stored at -80°C pending analysis. NAD⁺/NADH concentrations will be measured by Promega NAD/NADH-Glo assay kit (Promega, Southampton, UK). 2. Secondary outcomes: To measure physical performance (maximal hand grip strength) a Jamar Dynamometer will be used. As well as, to assess ATP/ADP concentrations and ratio, and respiratory chain deficiency will be assessed by histochemistry, and immunofluorescence (respectively). Mitochondrial DNA copy numbers will be assessed by quantitative PCR. Phosphocreatine recovery rate and NAD(P) changes will be measured by MR spectroscopy, and intramuscular fat infiltration by MRI (Magnetic resonance imaging). 3. Besides, Whole blood will be collected into EDTA tubes and frozen at -80C to test whether individuals with sarcopenia and low
---

	concentrations of NAD⁺/NADH ratios at biopsy, can be identified by less invasive biomarkers. 4. In the section on written informed consent, authors mention that blood will be taken for full blood count, urea, creatinine, and electrolytes, and liver function tests (bilirubin, alanine aminotransferase, and alkaline phosphatase). I think that information should be described in the methods section. ¿ Then how much blood sample will be required? Indicate the quantity. 5. However, I have a minor observation: the study period, as mentioned by the authors, indicates that “patient recruitment began in October 2022”, so ¿ Is the study already in progress? or the date needs to be update. This is an important point because the Journal states that: Manuscripts that report work already carried out will not be considered as protocols. 6. Medication adherence (collection/count). Although adherence to treatment is mentioned in the schedule of activities, this information is not in the protocol. I believe it should be described in the methods section. 5. Are research ethics (e.g. participant consent, ethics approval) addressed appropriately? Yes, the authors present an informed consent section. There, they explain that it will be obtained at home, clinic, or research center. The trial was approved by the UK Medicines and Healthcare Products Regulatory Agency (EuDRAC 2021-000993-28) UK Health Research Authority and Northeast -Tyne and Wear South Research Ethics Committee (IRAS 293565). Besides, it is registered at ISRCTN trial database (ISRCTN87404878). In addition, the authors point out that there is no conflict of interest between the sponsor (Newcastle Upon Tyne Hospitals NHS Foundation Trust, Freeman Hospital) and the development of the project, as they indicated on page 15, line 47: “The trial Sponsor has delegated responsibility for trial management to the CI, including trial design; collection, analysis, and interpretation of data; writing of the protocol publication and final clinical report”. Furthermore, the researchers point out that if side effects related to the intervention with acipimox occur, they will be recorded and their administration will be suspended, as indicated in the protocol: “Investigators may discontinue the trial treatment in the event of intolerable side effects occurring that are possibly, probably, or definitely related to trial medication or which constitute a serious adverse reaction (SAR) or suspected unexpected serious adverse reaction (SUSAR). Treatment will also be discontinued if the participant requests the medication to be withdrawn, requests not to undergo follow-up study measurements, or if creatinine clearance falls below 45 ml/min. If creatinine clearance falls from above 60ml/min to between 45 and 60ml/min during the study, the dose frequency will be reduced from three times a day to twice a day. If acipimox is discontinued this will automatically lead to withdrawal from the trial.” 6. Are the outcomes clearly defined? Yes, the outcomes are clearly defined, as explained above. Primary outcome: The change in skeletal muscle NAD⁺/NADH concentrations and the ratio between baseline and follow-up. To measure this outcome, a muscle biopsy of vastus lateralis muscle will be performed at baseline and follow-up.
--	---

	Secondary outcomes: 1) Physical performance, measured using a Jamar Dynamometer, 2) mitochondrial DNA copy number will be assessed via quantitative PCR, 3) Magnetic resonance spectroscopy will assess phosphocreatine recovery rate measured by ³¹P MRS of the calf, 4) Whole blood samples will be analyzed for NAD⁺/NADH concentrations and ratio using Q-NADMED Blood NAD⁺ and NADH assay kit; etc. 7. If statistics are used, are they appropriate and described fully? Yes, I consider that the analysis plan is appropriate, but the authors explained that a full statistical analysis plan will be developed during the trial, which will include an evaluation of the mechanistic outcomes data and correlation with the main study outcomes. Results will be analyzed using a paired t-test (if normally distributed) or a Wilcoxon signed-rank test (if not normally distributed). Binary secondary endpoints will be analyzed with a logistic regression model. Correlation between the changes in skeletal muscle NAD⁺/NADH ratios and changes in selected biomarkers (phosphocreatine recovery rate by MR spectroscopy, NAD⁺/NADH ratios in blood) will be performed using Pearson's correlation coefficient (where measures are normally distributed) or Spearman's rho (where measures are not normally distributed). 8. Are the references up-to-date and appropriate? Yes, the references are appropriate. Most of them are current, except for 7 (15%) references (references #9, #14, #15, #23, #27, #28, and #32). However, I consider that they are pioneering and classic studies in the interest of the protocol. 9. Do the results address the research question or objective? Does not apply, because the article type is a study protocol. 10. Are they presented clearly? Does not apply, because the article type is a study protocol. 10. Are the discussion and conclusions justified by the results. Does not apply, because the article type is a study protocol. 11. Are the study limitations discussed adequately? The lack of a control group could be a limitation of the study.
--	---

	12. Is the supplementary reporting complete (e.g. trial registration; funding details; CONSORT, STROBE or PRISMA checklist)? Yes, the supplementary reporting is complete. 13. To the best of your knowledge is the paper free from concerns over publication ethics (e.g. plagiarism, redundant publication, undeclared conflicts of interest)? Yes, to my knowledge, it is an original research protocol.
--	---

VERSION 1 – AUTHOR RESPONSE

Reviewer: 1 Dr. Philip Chilibeck, University of Saskatchewan

Introduction: I suggest providing details on how sarcopenia is measured in the introduction. There is controversy on the most adequate way to measure this.

- Thank you for your advice. We have added clarification regarding the definition of sarcopenia referenced in the study. (page 5)

Mitochondrial dysfunction is associated with sarcopenia. Could the authors better describe the link in the introduction – is it due to excessive production of reactive oxygen species which induce damage to cellular membranes?

- We have added further detail on the relationship between mitochondrial oxidative capacity, muscle strength.
- We have highlighted the numerous mitochondrial changes associated with muscle atrophy.
- We have also as the author suggested highlighted the role of ROS in damage to local cellular components. (page 6)

What are the usual adverse events associated with acipimox?

- The common adverse effects associated with acipimox as reported in summary of product characteristics have been added to the manuscript (page 9)

Page 8, Interventions: Please include the length of the intervention in this section. I believe this is three weeks, according to the abstract.

- Thank you for highlighting this omission. This has been added the manuscript at the end of the intervention section. (page 9)

Please provide justification for this length of intervention. Is this long enough to show significant changes in your primary and secondary outcomes?

- We have now added some additional information on this. We based our choice of dose and duration on a previous study of acipimox in patients with type 2 diabetes mellitus. (page 9)

Is there any chance that co-administration of aspirin might affect some of your outcomes as NSAIDs might affect muscle mass and strength (please see: <https://pubmed.ncbi.nlm.nih.gov/21160058/>)?

- Inhibiting low grade inflammation with aspirin or other COX inhibitors has been mooted as a possible intervention to delay the onset or slow the progression of sarcopenia, although without good evidence to date. Co-administration of aspirin should improve tolerability of acipimox and thus reflects the way that the intervention might be deployed in clinical practice. It does make it more difficult to dissect out the mechanistic underpinnings of any improvement in muscle function however. We have now acknowledged this in the strengths and limitations section; measurement of NAD-specific outcomes provides some mitigation against this. (page 4)

Page 12: Sample size: It is indicated that a projected change of 1 SD was used in the sample size calculation. Please indicate the exact number that was used in the sample size calculation (i.e., the actual NAD+/NADH that corresponds to 1 SD).

- As this is a feasibility study, few data exist on which to base a sample size calculation. The sample size has been selected to enable detection of a 1 SD change in measures of mitochondrial NAD levels, ATP levels or respiratory chain function. To detect this change with an alpha of 0.05 and 80% power requires 11 paired observations (using a paired t-test). We plan to recruit 16 participants to allow for dropout and non-completion of the course of medication."

Page 14: Were patient-desired outcomes actually the biopsy and P-MRS outcomes? I would suspect patients might not understand mechanistic outcomes and might place more importance on functional outcomes (i.e., strength, gait speed).

- Patients do place more importance on measures of physical performance such as strength and gait speed, but these were not the focus of this early-phase experimental medicine study. We did not ask patients whether they would have preferred functional outcomes; instead we asked if the mechanistic outcomes we proposed were likely to be acceptable to patients. A larger trial will be needed to assess patient centred outcomes such as physical performance and quality of life (which would only be done if this initial study had favourable results) and we have now clarified this on page 16.

Reviewer: 2

Dr. Lourdes Barbosa-Cortés, Instituto Mexicano del Seguro Social Unidad de Investigación Médica en Nutrición

In the section on written informed consent, authors mention that blood will be taken for full blood count, urea, creatinine, and electrolytes, and liver function tests (bilirubin, alanine aminotransferase, and alkaline phosphatase). I think that information should be described in the methods section. ¿Then how much blood sample will be required? Indicate the quantity.

- We have placed these details in the "screening" portion of the manuscript as these bloods were to assess eligibility and safety for inclusion in the trial rather than as an outcome measure. All tests that are measured as a primary or secondary outcome are mentioned elsewhere.
- We have now clarified this section and included volume of blood sample required. (page 14)

However, I have a minor observation: the study period, as mentioned by the authors, indicates that "patient recruitment began in October 2022", so ¿Is the study already in progress? or the date needs to be update. This is an important point because the Journal states that: Manuscripts that report work already carried out will not be considered as protocols.

- The study started in October 2022 we submitted paper in June 2022 and were only in the very early stages of collecting data (4 participants enrolled). No data analysis has begun at the time of writing. This was highlighted to the editorial team who based in this information were willing to consider the manuscript. The study is ongoing.

Medication adherence (collection/count). Although adherence to treatment is mentioned in the schedule of activities, this information is not in the protocol. I believe it should be described in the methods section.

- Thank you for highlighting this oversight. A section has now been added to the manuscript. (page 11)

Are the study limitations discussed adequately? The lack of a control group could be a limitation of the study.

- This has now been highlighted in the limitations (page 4)

VERSION 2 – REVIEW

REVIEWER	Chilibeck, Philip University of Saskatchewan
REVIEW RETURNED	27-Dec-2023
GENERAL COMMENTS	Thanks for responding to my previous comments

VERSION 2 – AUTHOR RESPONSE